# Fully Automated Colorimetric Analysis of the Optic Nerve Aided by Deep Learning and Its Association with Perimetry and OCT for the Study of Glaucoma

**DOI:** 10.3390/jcm10153231

**Published:** 2021-07-22

**Authors:** Marta Gonzalez-Hernandez, Daniel Gonzalez-Hernandez, Daniel Perez-Barbudo, Paloma Rodriguez-Esteve, Nisamar Betancor-Caro, Manuel Gonzalez de la Rosa

**Affiliations:** 1INSOFT S.L., 25 de Julio, 34, 38004 Santa Cruz de Tenerife, Spain; martaglezhdez@gmail.com (M.G.-H.); management@insoft.es (D.G.-H.); danieltf@gmail.com (D.P.-B.); 2Ophthalmology Department, Hospital Universitario de Canarias, Carretera Ofra s/n, 38320 San Cristobal de La Laguna, Spain; 3Facultad de Ciencias de la Salud, Universidad de La Laguna, C/Sta, María Soledad s/n, 38200 San Cristobal de La Laguna, Spain; paloma.rguezesteve@gmail.com (P.R.-E.); nisamar89@hotmail.com (N.B.-C.)

**Keywords:** glaucoma, deep learning, perimetry, optic nerve

## Abstract

Background: Laguna-ONhE is an application for the colorimetric analysis of optic nerve images, which topographically assesses the cup and the presence of haemoglobin. Its latest version has been fully automated with five deep learning models. In this paper, perimetry in combination with Laguna-ONhE or Cirrus-OCT was evaluated. Methods: The morphology and perfusion estimated by Laguna ONhE were compiled into a “Globin Distribution Function” (GDF). Visual field irregularity was measured with the usual pattern standard deviation (PSD) and the threshold coefficient of variation (TCV), which analyses its harmony without taking into account age-corrected values. In total, 477 normal eyes, 235 confirmed, and 98 suspected glaucoma cases were examined with Cirrus-OCT and different fundus cameras and perimeters. Results: The best Receiver Operating Characteristic (ROC) analysis results for confirmed and suspected glaucoma were obtained with the combination of GDF and TCV (AUC: 0.995 and 0.935, respectively. Sensitivities: 94.5% and 45.9%, respectively, for 99% specificity). The best combination of OCT and perimetry was obtained with the vertical cup/disc ratio and PSD (AUC: 0.988 and 0.847, respectively. Sensitivities: 84.7% and 18.4%, respectively, for 99% specificity). Conclusion: Using Laguna ONhE, morphology, perfusion, and function can be mutually enhanced with the methods described for the purpose of glaucoma assessment, providing early sensitivity.

## 1. Introduction

Glaucoma is a disease with a relatively high incidence, estimated between 2.09% and 5.82% of the adult population [1]. Its diagnosis in early stages is difficult due to its almost asymptomatic onset and controversy of criteria regarding its initial signs.

Intraocular pressure is an important pathogenesis factor; while not the only one, it is currently the only one that can be targeted for treatment. Many other factors seem to have an influence on the disease: genetic, morphological, vasospasm, intracranial pressure, tissue thickness and morphology, sleep apnoea, etc.

For many years, visual field assessment and optic nerve head examination were the main diagnostic procedures. More recently, morphological tests have been proposed, and many authors have argued their greater precocity [2], although, as shown in this paper, this opinion is debatable. Examples are the scanning laser polarimeter (GDx), the Heidelberg retina tomograph (HRT) and the optical coherence tomograph (OCT), which is currently widely used. More recently, OCT angiographies have been introduced to analyse vascularization [3], especially in the retinal areas surrounding the optic nerve.

We developed a simple colorimetric procedure to estimate the presence of haemoglobin and its distribution on the optic disc, which reflects its perfusion and morphology, using conventional colour retinographies (Laguna ONhE) [4]. The method has been shown to have good sensitivity and specificity in previous studies [5,6,7,8,9,10,11,12]. However, deep learning methods for the segmentation of optic nerve edges were later incorporated to facilitate its use and improve its reproducibility [13].

Artificial intelligence, especially convolutional networks, allow the experience of experts to be transferred with great precision to other possible users in different medical specialties, most notably in ophthalmology [14]. Transfer Learning’s new procedures simplify the development work, taking advantage of the previous training of networks on generic problems [15].

In a previous experiment [13], we used a deep learning U-Net architecture for semantic segmentation [16]. In this case, 40,000 images were used to segment the optic disc, identifying the inner edge of Elschnig’s scleral ring.

The importance of perfusion in glaucoma is well recognised [3]. A recent paper using this same version of the algorithm showed that its results are in the range of an OCT angiography [17]. The aim of this paper was to take advantage of these new automatic procedures based on experience to facilitate the use of the application and to improve its reproducibility, sensitivity, and specificity. In particular, an attempt was made to combine the morphological and perfusion information provided by the method with functional perimetric data representative of visual field homogeneity [18] to assess whether this provided robustness in the diagnostic decision.

## 2. Materials and Methods

The study protocol of this cross-sectional retrospective study adhered to the principles of the 1975 Declaration of Helsinki revised in 2013, and was approved by the Research Ethics Committee of the Hospital Universitario de Canarias (CHUC_2018_09 (V3)). Consent was obtained from all participants.

### 2.1. Automation of the Laguna ONhE Method for Estimation of Haemoglobin in the Optic Nerve Head

The presence of haemoglobin in the tissue was estimated by Laguna ONhE with reference to the colour of the vessels. Haemoglobin mainly absorbs green radiation and reflects most of the red. Therefore, the reference colour of the vessels was calculated using the values of the red (R) and green (G) channels of their pixels, to which the formula (R−G)/R is applied. The same equation was used for the pixels of the tissue, and finally, the result was expressed as a percentage. An estimate of cup size and position was also obtained, and the results of the cup, rim sectors, vertical cup/disc ratio (Hb-C/D) and cup/disc area ratio were compared with the percentiles achieved in the normal population [10]. Each fundus camera model was calibrated to achieve an equivalent response.

To achieve full automation of the method, five neural networks were used: one to segment the edges of the optic nerve already described [13], one for vessel segmentation using 4195 optic disc images, one to identify the eye as left or right using 4201 images, one to recognize image quality using 7048 images, and one to identify between normality (using 1518 images) and confirmed or suspected glaucoma (using 1596 images). The technical method is described in detail in the Appendix A “Computing development setup”.

The classification results obtained by deep learning were associated with the distribution of haemoglobin and the estimated Cup/Disc ratios to define a new value for the “Globin Distribution Function” (GDF) index, as previously described [4,5]. Once the value of the deep learning classifier was normalized to the mean values and standard deviation in the normal population, it influenced the result of the current GDF index by 45%. In the remaining 55%, the rest of the usual variables that we used in previous studies also intervened with normalized values. An example of the graphic results is shown in Figure 1. It shows how the method automatically performs the analysis of a retinography (a). Once its quality has been checked, the position of the optic nerve is identified, the inner edge of Elschnig’s scleral ring is defined, which is more internal than the apparent clear edge, and the size and shape of the cup is estimated (b). Its veins and arteries are then segmented, the colour of which is used to estimate the relative haemoglobin of the rest of the nerve tissue, shown in a colour code (c). In each sector of the optic disc, cup or rim, their areas are estimated as a percentage of the total disk area and expressed in colour if it corresponds to what is expected in a normal optic nerve (d).

### 2.2. Combination of Laguna ONhE and Perimetric Indices

The pattern standard deviation (PSD) perimetric index and its equivalent in the Octopus perimeters, the square root of loss variance (sLV), are well known. In essence, the pointwise differences between the patient’s sensitivity thresholds and the expected value in a subject of the same age were calculated, and the standard deviation of all values was obtained. In general, these values increase as the disease progresses, but in advanced defects they reduce as the number of points with no detectable sensitivity (0 dB) increases. To achieve a linear response, in advanced glaucoma, a modification was made, as previously described [19], taking into account the mean defect/deviation (MD), using Equation (1).

Equation (1): sLV modification to provide a linear response.
(1)$$\mathrm{If}\;\mathrm{abs}\left(\mathrm{MD}\right)>16.33,\;\mathrm{then}\;\mathrm{sLV}=\mathrm{sLV}+\frac{\mathrm{abs}\left(\mathrm{MD}\right)-16.33}{0.84}$$

The threshold coefficient of variation (TCV) is an index of regularity described by our research group, which represents the harmony of one’s own visual field, without comparing it with patterns of normality [18]. A characteristic for normality is stability or harmony, and for pathology, is irregularity or fluctuation. We aimed to analyse them in the subject themself, regardless of whether their thresholds are close or far from the normal average, in the same way that a subject can be harmonious and of short stature or harmonious and tall. This way, TCV is a less complex index than PSD, as it is independent of the age of the subject. In normal subjects and initial and moderate glaucoma, it is calculated by dividing the standard deviation of the thresholds in 16 symmetrical positions of the visual field by their mean value, and multiplying by 100. In advanced glaucoma, where absolute scotomas are present, both values are adjusted to their number.

### 2.3. Datasets for the Laguna ONhE, OCT and Perimetric Indices

Two groups were included, consisting of 477 healthy eyes from 409 subjects and 333 eyes with confirmed or suspected chronic open-angle glaucoma from 246 subjects. Healthy subjects consisted of hospital staff, patient relatives, or people who needed refraction but did not have eye abnormalities.

All cases had corrected visual acuity of 20/40 or higher, refractive error with spherical equivalent of less than ±5.00 dioptres, astigmatism less than ±2 dioptres, and open anterior chamber angle. Subjects with cataracts reaching the specified visual acuity were not excluded. Previous cataract or glaucoma surgery were also not exclusion criteria. Patients with associated ocular diseases which could interfere with the interpretation of the results, such as optic neuritis, coloboma, and papillary oedema, were not included. All the participants underwent a complete examination, including visual acuity, slit lamp examination, Goldmann tonometry, and fundus examination, within a maximum time interval of one week.

All cases were examined with the Cirrus spectral-domain optical coherence tomograph (OCT; Carl Zeiss Meditec, Jena, Germany), using the optic disc cube 200 × 200 acquisition protocol (software version 5.2). All images were acquired with a quality greater than 6/10.

Three fundus cameras, two perimeters, and two different perimetric strategies were used to verify that the results of the evaluated method did not depend on the instruments used. In total, 213 normal cases and 110 glaucomas were examined with a Nidek AFC-210 non-mydriatic fundus retinograph (Nidek Co., LTD, Aichi, Japan) and a white-on-white Spark strategy in an Easyfield perimeter (Oculus Optikgeräte GmbH, Wetzlar, Germany) [20]. Then, 87 normal cases and 70 glaucomas were examined with a Kowa Wx non-mydriatic fundus retinograph (Kowa Co. Ltd., Tokyo, Japan) and with the Easyfield perimeter using Spark strategy, and 177 normal and 153 glaucomas were examined with a Horus Scope DEC-200 handheld fundus camera (MiiS, Hsinchu, Taiwan) and an Octopus 300 perimeter (Hagg-Streit AG, Bern, Switzerland) using the Tendency Oriented Perimetry (TOP) strategy [21].

All control and glaucoma patients underwent perimetric assessment, having undergone at least two previous examinations. Healthy eyes had intraocular pressure less than 21 mmHg, and no abnormal results in the visual field or on the optic disc. The “glaucoma” group comprised glaucoma and glaucoma-suspected eyes. Not all patients in the “glaucoma” group had defects characteristic of the optic nerve or visual field. In some cases, there were only signs of suspicion, such as intraocular pressure greater than 21 mmHg, associated with a family history of glaucoma, an optic disc with a dubious appearance, or borderline visual fields, such as a mean deviation exceeding −2 dB or points outside normal limits on the defect curve. Subjects with ocular pressures greater than 25 mmHg, or pressures between 21 and 25 mmHg accompanied by a thin cornea (less than 500 μm), were also included.

Two types of analysis were carried out. In the first, no strict boundary was established between confirmed and suspected glaucoma, so as not to introduce an a priori criterion that could alter an objective interpretation of the results [22]. This methodology was widely discussed in a previous paper [23]. Additionally, two groups were analysed separately (confirmed glaucoma and glaucoma suspects): confirmed glaucoma was defined based on the presence of glaucomatous visual field loss in standard automated perimetry (pattern standard deviation or mean deviation of <5%) and signs of glaucomatous neuropathy (Laguna ONhE GDF or OCT-rim area of <5%). Those who did not meet these criteria, but met some of the criteria described in the previous paragraph, were considered as glaucoma suspects.

### 2.4. Statistical Analysis

Clinical statistical analyses were performed using the Excel 2016 program (Excel, Microsoft Corp., Redmond, WA, USA) and MedCalc (Version 18.9–64 bits; MedCalc software bvba, Mariakerke, Belgium). The statistical comparison between the results of the different AUCs were calculated in MedCalc using the criteria described by DeLonng et al. [24].

For statistical purposes, the sign of MD values of the Octopus Perimeter was inverted, to use the same criterion in both perimeters.

In order to associate different indices, they were previously normalized in relation to the mean value and standard deviation of all normal cases.

## 3. Results

The reference group was composed of 167 male eyes and 310 female eyes, aged 44.56 ± 13.23 years, with a perimetric MD of 0.18 ± 1.56 dB and PSD-sLV of 1.48 ± 0.52 dB. The rim area of the Cirrus-OCT was 1.42 ± 0.29 mm^2^, the retinal nerve fibre layer thickness (RNFLT) was 91.67 ± 9.77 µm, and the vertical cup/disc ratio (OCT-C/D) was 0.45 ± 0.15.

The glaucoma group consisted of 176 male eyes and 157 female eyes, aged 63.63 ± 11.74 years, with a perimetric MD of −8.85 ± 8.45 dB and PSD-sLV of 4.43 ± 2.82 dB. The rim area was 0.77 ± 0.34 mm^2^, the RNFLT was 68.75 ± 16.44 µm, and the OCT-C/D was 0.74 ± 0.14. Differences were statistically significant in all cases (*p* < 0.0001). Among them, 235 met the criteria for confirmed glaucoma, and 98 were considered to be suspected glaucoma. Glaucoma suspects had an MD of −0.92 ± 2.71 dB, and confirmed glaucoma had an MD of −12.16 ± 7.81 dB.

### 3.1. Results of the Indices of the Three Testing Methods on the Total Sample (without Separating Confirmed and Suspected Glaucoma)

The GDF index result in the groups of validation datasets is shown in Table 1. The Laguna ONhE GDF index obtained significantly higher results than all perimetric and Cirrus-OCT indices. The best OCT index was rim area and the best perimetric index TCV, with no significant differences between them.

Figure 2 shows the receiver operating characteristic (ROC) curves obtained for the Laguna ONhE GDF index, the OCT indices, and the perimetric indices. To improve the identification of differences, the upper left part of the curves is enlarged. GDF presented a sensitivity higher than the rim area for 95.0% (*p* = 0.0121) and 99.0% specificities (*p* = 0.0131), which was also higher than the rest of the perimetric indices and OCT (*p* < 0.0001).

### 3.2. Combination of Laguna ONhE and Perimetric Indices on the Total Sample (without Separating Confirmed and Suspected Glaucoma)

Table 2 shows the results of the combinations of the two examination methods (Laguna ONhE and OCT) and the perimetric indices of uniformity and harmony. The AUCs did not show significant differences when combining GDF with each of the two perimetric indices, but these two combinations produced significantly larger curves than the combinations of perimetric and OCT indices. The RNFLT and PSDr combination performed better than the other OCT indexes combinations with the perimetric PSDr. The combinations with the perimetric MD performed worse and were thus ignored.

Figure 3 shows the ROC curves obtained for the combinations of the Laguna ONhE and OCT indices with those of perimetry. The combinations of GDF and TCV and GDF and rectified PSD-sLV did not present differences in sensitivity for 95.0% (*p* = 0.790) and 99.0% (*p* = 0.674) specificities. For 95.0% specificity, both combinations had higher sensitivities than the combinations of perimetry and OCT (*p* < 0.0005). For 99.0% specificity, the combination GDF and TCV had higher sensitivity than the rim area and rectified PSD-sLV (*p* = 0.011). GDF and rectified PSD-sLV surpassed this with *p* = 0.032. Both combinations surpassed the combinations used by the other OCT and perimetrics indices (*p* < 0.0005).

A linear relationship was observed between perimetric MD and the GDF and TCV combination (Figure 4a), while the relationship was largely curvilinear when comparing MD with the combination of OCT morphological indices and TCV or PSDr (Figure 4b). The correlation between MD and GDF and TCV (r = 0.9132) was significantly higher than that between MD and GDF and PSDr (r = 0.8927) (*p* = 0.0252).

### 3.3. Results of the Indices of the Three Testing Methods on the Total Sample Compared to Confirmed and to Suspected Glaucoma

Table 3 shows that Laguna ONhE GDF index obtained significantly higher results than all perimetric and Cirrus-OCT indices in suspect glaucomas. It also achieves the highest ROC area in confirmed glaucoma, but without significant differences with several of the other indices.

### 3.4. Performance of Combined Indexes in Normal Subjects as Compared to Confirmed and to Suspected glaucoma

Table 4 shows the results of the best combinations of two non-psychophysical examination methods (Laguna ONhE and OCT), with the perimetric indices of uniformity and harmony in confirmed and suspected glaucoma. Significant differences between the two could be observed in glaucoma suspects. In suspected glaucomas, the combinations of the three OCT indices and the perimetric PSDr did not show significant differences. The combinations with the perimetric MD were worse and thus ignored.

## 4. Discussion

Several current papers that apply deep learning to the study of glaucoma exhibit interesting theoretical approaches; however, they are not always based on real-life scenarios, but on samples collected by outside groups. Several review papers [14,25] highlight the results of others that exclude suspected glaucomas from the sample, not fully reflecting the reality of the clinical problem. They often do not clarify the type of sample analysed [26], or they base the reference classification (normal or glaucoma) exclusively on subjective expert opinions on the images [27]. Others include only cases with large cups as glaucomas [28], or do not indicate the specificity achieved [29]. Achieving a good sensitivity is not sufficient in this type of task [15] if the specificity is not high. Our analysis was performed on a real sample, without excluding doubtful or intermediate cases, and trying to avoid biases that may occur when differentiating between confirmed and suspected glaucoma [23], although both groups were analysed separately, according to the most common criteria in the literature.

Selectively analysing the optic nerve and controlling image quality has undoubted advantages over the use of wider eye fundus images [30], although it has been suggested that the sectoral atrophy of fibres could enrich the diagnosis [31].

Attempting to emulate the behaviour of an OCT to obtain its most efficient indices is an interesting approach [32]. However, the classifiers obtained through deep learning produce results that tend to involve the extremes of a dichotomous series (for example: Yes/No glaucoma or Yes/No image of adequate quality) but do not provide a gradual value of the degree of defect. For this reason, in our case, we attempted to combine these results with the estimation of other indices, such as the cup/disc ratio, which is obtained by multiple regression from sectoral haemoglobin estimates [10], and/or with the results of visual field examinations. This type of combination provides new indices with progressive gradation to better assess the level of defect and, likely, its degree of progression.

In general, published procedures require the intervention of experts for pre-processing or assessing if the images are of sufficient quality for analysis. Our procedure reached a degree of full automation in this respect, requiring no human intervention.

It is well known that the relationship between morphology and function estimated by OCT and perimetry is not linear. However, the precocity of detectable morphological damage with respect to functional damage has been questioned by several authors [33,34]. Some aspects can be complementary. Indeed, our study suggests that the association between the Laguna ONhE algorithm and perimetry seems to provide highly promising results. In particular, the linearity between the perimetric mean deviation (MD), the combination of the morphological and perfusion information presented by the Laguna ONhE GDF index, and the functional disharmony assessed by TCV provides a new perspective in the interpretation of these relationships.

It is noteworthy that this combination achieves high sensitivity while retaining high specificity. In a disease of uncertain onset, such as glaucoma, it is advisable to evaluate it with highly specific procedures, in order not to establish unnecessary treatments and to avoid overloading the health systems.

The combination of the indices provided by OCT angiography with perimetric indices should be evaluated in the future, in order to compare their results with those presented in this paper. However, an important factor to take into account in this type of comparison is the cost–benefit ratio for healthcare systems, given the low cost of some current fundus cameras, and the possibility of evaluating their images via telemedicine, without additional equipment. Such an evaluation should also be carried out in the near future.

## 5. Conclusions

The Laguna ONhE method arose from the observation of differences in RGB frequency histograms of optic disc structures (vessels, rim, and cup) when analysing colour photographs of the optic nerve, as can be seen in Figure 6 of our first publication [4]. The favourable results of a simple, non-invasive test, such as the one we propose for the assessment of glaucoma, were cited in the introduction, and are further evident in this new study, as they are enhanced by functional perimetric results. The application of deep learning facilitates the use of the Laguna ONhE software, taking advantage of expert experience, and improving its reproducibility, sensitivity, and specificity. Morphology, function, and perfusion can be combined for the optimal evaluation of glaucoma. However, these results should be confirmed by other independent studies.

## Figures and Tables

**Figure 1 jcm-10-03231-f001:**
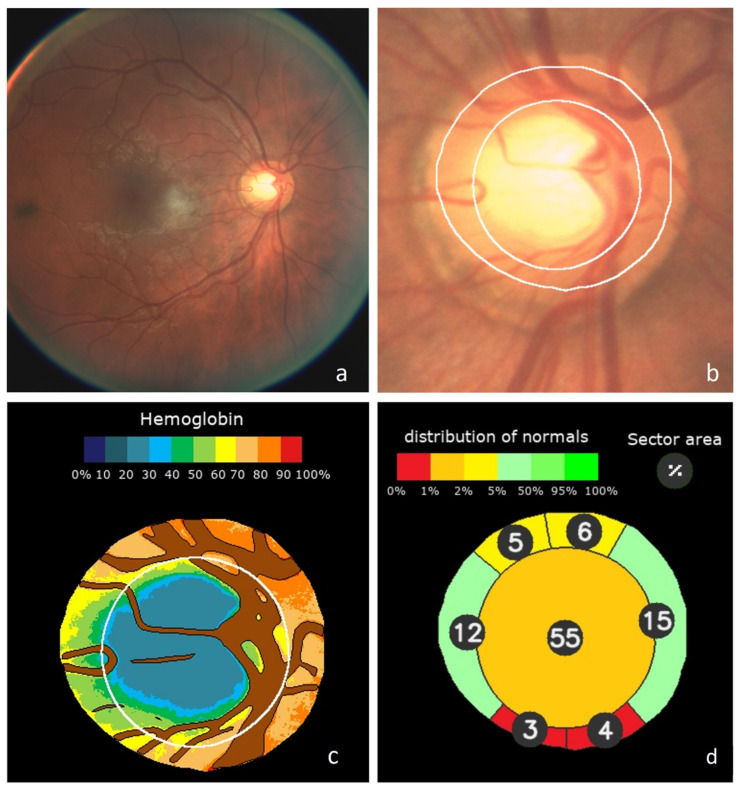
Example of Laguna ONhE analysis: (**a**) Original wide-field eye fundus image. (**b**) Identification of the optic disc and boundaries segmentation. Central cup estimated from the haemoglobin distribution. (**c**) Segmentation of reference vessels and pseudo-colour image of haemoglobin distribution. (**d**) Estimated sector areas as a percentage of the total area, and compared to a normal reference population.

**Figure 2 jcm-10-03231-f002:**
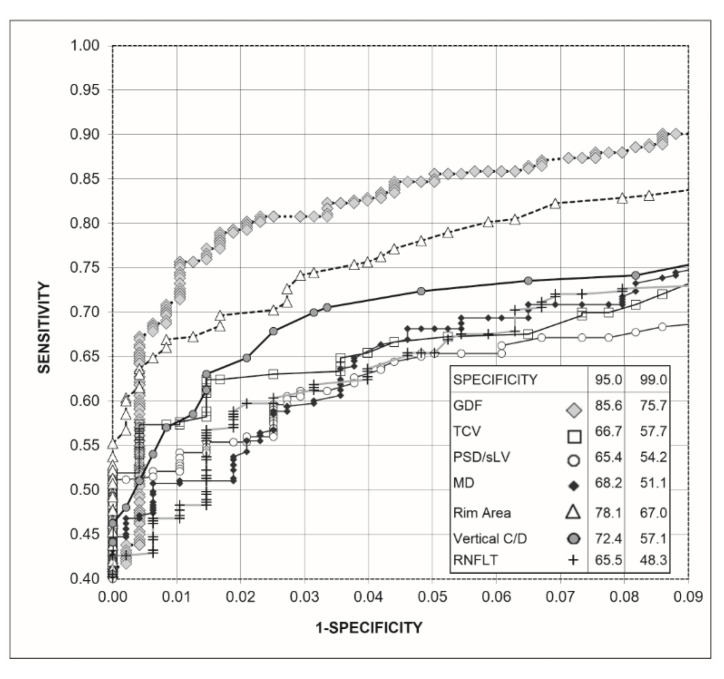
Receiver operating characteristic curve (ROC) obtained with Laguna ONhE (GDF), Cirrus-OCT (rim area, vertical C/D ratio, and RNFLT) and perimetry (MD, TCV, and PSD-sLV) indices, as well as percentage sensitivities for 95% and 99% specificities.

**Figure 3 jcm-10-03231-f003:**
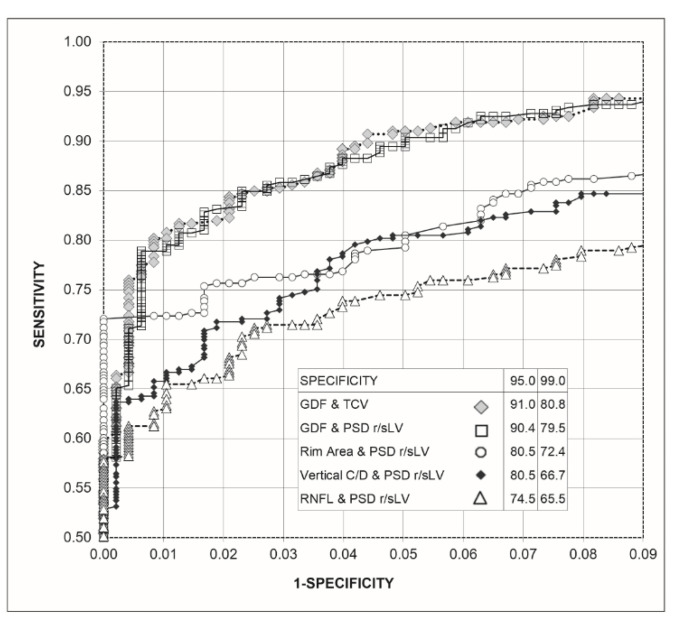
Graph of the receiver operating characteristic curve (ROC) of the combinations of Laguna ONhE and Cirrus-OCT indices with the perimetrics, as well as percentage sensitivities for 95% and 99% specificities.

**Figure 4 jcm-10-03231-f004:**
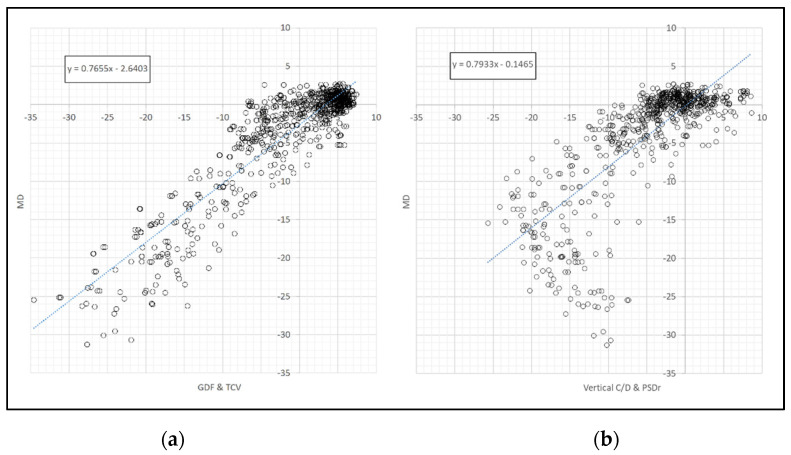
Relationship between the perimetric MD index and (**a**) the combined GDF and TCV and (**b**) the vertical C/D and PSDr (right) indices.

**Table 1 jcm-10-03231-t001:** Performance of Laguna OnhE, OCT, and perimetry indices.

	GDF	RNFLT	Rim Area	OCT-C/D	MD	PSDr	TCV
AUC	0.970	0.875	0.926	0.921	0.897	0.883	0.904
SE	0.0054	0.0135	0.0104	0.0099	0.012	0.013	0.0115
CI	0.956–0.981	0.850–0.897	0.906–0.943	0.901–0.939	0.874–0.917	0.859–0.904	0.881–0.923
	**GDF**	**RNFLT**	**Rim Area**	**OCT-C/D**	**MD**	**PSD-sLVr**	
RNFLT	*p* < 0.0001						
Rim Area	*p* < 0.0001	*p* < 0.0001					
OCT-C/D	*p* < 0.0001	*p* = 0.0006	*p* = 0.5812				
MD	*p* < 0.0001	*p* = 0.1389	*p* = 0.0422	*p* = 0.0869			
PSDr	*p* < 0.0001	*p* = 0.6013	*p* = 0.0039	*p* = 0.0083	*p* = 0.0692		
TCV	*p* < 0.0001	*p* < 0.0001	*p* = 0.0824	*p* = 0.1731	*p* = 0.5053	*p* = 0.0066	

GDF = Laguna OnhE globin distribution factor; RNFLT = OCT retina nerve fiver layer thickness; OCT-C/D = OCT vertical cup/disc ratio; MD = perimetric mean defect or deviation; PSDr = perimetric pattern standard deviation or square root of loss variance rectified; TCV = perimetric threshold coefficient of variation; AUC = area under the receiver operating characteristic curve; SE = standard error; and CI = 5–95% confidence intervals.

**Table 2 jcm-10-03231-t002:** Performance of the combination of Laguna ONhE and Cirrus-OCT variables and perimetric variables.

	GDF and TCV	GDF and PSDr	Rim Area and PSDr	OCT-C/D and PSDr	RNFLT and PSDr
AUC	0.978	0.976	0.945	0.947	0.915
SE	0.00459	0.00506	0.00872	0.0079	0.0109
CI	0.965–0.987	0.963–0.986	0.927–0.960	0.929–0.961	0.894–0.933
	**GDF and TCV**	**GDF and PSDr**	**Rim Area and PSDr**	**OCT-C/D and PSDr**	
GDF and PSDr	*p* = 0.2974				
Rim Area and PSDr	*p* = 0.0001	*p* = 0.0002			
OCT-C/D and PSDr	*p* < 0.0001	*p* < 0.0001	*p* = 0.8728		
RNFLT and PSDr	*p* < 0.0001	*p* < 0.0001	*p* = 0.0015	*p* = 0.0004	

GDF = Laguna ONhE globin distribution factor; RNFLT = OCT retina nerve fiver layer thickness; OCT-C/D = OCT vertical cup/disc ratio; MD = perimetric mean defect or deviation; PSDr = perimetric pattern standard deviation or square root of loss variance rectified; TCV = perimetric threshold coefficient of variation; AUC = area under the receiver operating characteristic curve; SE = standard error; and CI = 5–95% confidence intervals.

**Table 3 jcm-10-03231-t003:** Performance of Laguna ONhE, OCT, and perimetry indices.

	GDF	RNFLT	Rim Area	OCT-C/D	MD	PSDr	TCV
AUC susp.	0.932	0.709	0.827	0.834	0.687	0.651	0.718
AUC conf.	0.986	0.944	0.968	0.958	0.985	0.98	0.981
SE susp.	0.013	0.0303	0.025	0.0223	0.0297	0.0311	0.029
SE conf.	0.00437	0.0107	0.00877	0.00867	0.00316	0.00432	0.00391
CI susp.	0.908–0.951	0.670–0.746	0.794–0.857	0.801–0.864	0.647–0.725	0.611–0.690	0.679–0.754
CI conf.	0.974–0.993	0.925–0.960	0.952–0.979	0.940–0.971	0.973–0.993	0.966–0.989	0.968–0.990
	**GDF**	**RNFLT**	**Rim Area**	**OCT-C/D**	**MD**	**PSDr**	
RNFLT susp.	*p* < 0.0001						
RNFLT conf.	*p* = 0.0002						
RimArea susp.	*p* < 0.0001	*p* = 0.0002					
Rim Area conf.	*p* = 0.0635	*p* = 0.0490					
OCT-C/D susp.	*p* < 0.0001	*p* = 0.0001	*p* = 0.7755				
OCT-C/D conf.	*p* = 0.0019	*p* = 0.2691	*p* = 0.1642				
MD susp.	*p* < 0.0001	*p* = 0.5917	*p* = 0.0004	*p* = 0.0001			
MD conf.	*p* = 0.8630	*p* = 0.0002	*p* = 0.0636	*p* = 0.0025			
PSDr susp.	*p* < 0.0001	*p* = 0.1429	*p* < 0.0001	*p* < 0.0001	*p* = 0.2399		
PSDr conf.	*p* = 0.2847	*p* = 0.0015	*p* = 0.2264	*p* = 0.0199	*p* = 0.2146		
TCV susp.	*p* < 0.0001	*p* = 0.8057	*p* = 0.0022	*p* = 0.0008	*p* = 0.2796	*p* = 0.0050	
TCV conf.	*p* = 0.3720	*p* = 0.0008	*p* = 0.1663	*p* = 0.0106	*p* = 0.2885	*p* = 0.6232	

GDF = Laguna ONhE globin distribution factor; RNFLT = OCT retina nerve fiver layer thickness; OCT-C/D = OCT vertical cup/disc ratio; MD = perimetric mean defect or deviation; PSDr = perimetric pattern standard deviation or square root of loss variance rectified; TCV = perimetric threshold coefficient of variation; AUC = area under the receiver operating characteristic curve; SE = standard error; and CI = 5–95% confidence intervals.

**Table 4 jcm-10-03231-t004:** Performance of the combination of Laguna ONhE and Cirrus-OCT variables and perimetric variables between reference and confirmed (conf.) and suspect (susp.) glaucoma cases.

	GDF and TCV	GDF and PSDr	Rim Area and PSDr	OCT-C/D and PSDr	RNFLT and PSDr
AUC conf.	0.995	0.995	0.988	0.988	0.987
AUC susp	0.935	0.932	0.843	0.847	0.742
SE conf.	0.0021	0.0021	0.0049	0.0035	0.0033
SE susp.	0.013	0.015	0.024	0.022	0.029
CI conf.	0.987–0.999	0.987–0.999	0.977–0.995	0.977–0.995	0.976–0.994
CI susp	0.912–0.954	0.908–0.951	0.810–0.871	0.815–0.876	0.704–0.77
	**GDF and TCV**	**GDF and PSDr**	**Rim Area and PSDr**	**OCT-C/D and PSDr**	
GDF and PSDr conf.	*p* = 0.695				
GDF and PSDr susp.	*p* = 0.323				
Rim Area and PSDr conf.	*p* = 0.187	*p* = 0.195			
Rim Area and PSDr susp.	*p* < 0.0001	*p* = 0.0002			
OCT-C/D and PSDr conf.	*p* = 0.060	*p* = 0.064	*p* = 0.8728		
OCT-C/D and PSDr susp.	*p* < 0.0001	*p* < 0.0001	*p* = 0.839		
RNFLT and PSDr conf.	*p* = 0.036	*p* = 0.037	*p* = 0.852	*p* = 0.859	
RNFLT and PSDr susp.	*p* < 0.0001	*p* < 0.0001	*p* = 0.0005	*p* = 0.0001	

GDF = Laguna ONhE globin distribution factor; RNFLT = OCT retina nerve fiver layer thickness; OCT-C/D = OCT vertical cup/disc ratio; MD = perimetric mean defect or deviation; PSDr = perimetric pattern standard deviation or square root of loss variance rectified; TCV = perimetric threshold coefficient of variation; AUC = area under the receiver operating characteristic curve; SE = standard error; and CI = 5–95% confidence intervals.

## Data Availability

The datasets analysed and generated during the study can be found at the following link: https://cloud.insoft.es/s/sRaFgmjQp7S4sgR.

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
