# Peer review of "Fully Automated Colorimetric Analysis of the Optic Nerve Aided by Deep Learning and Its Association with Perimetry and OCT for the Study of Glaucoma"

_jcm, 2021, doi:10.3390/jcm10153231_

Round 1
Reviewer 1 Report
The authors present an interesting study about the diagnostic capacity of a topographic and colorimetric analysis system of the optic nerve and its comparison with other functional and structural analysis tools.
Some minor suggestions:
The title only refers to the visual field and not to the OCT that is also included in the study.
Line 40. Optic nerve head examination has also been used for many years before other structural analysis were introduced.
Line 62. OCT angiography
Line 96. Remaining 55%
Line 98. It would be interesting to know more about how the inner edge of Elschnig's scleral ring is delimited.
Line 167. The classification criteria for the groups suspected of glaucoma and confirmed glaucoma are not clear in the text. Please clarify to confirm that there is no selection bias. A table with the different criteria used would be useful for readers.
Line 287. It should be section 3.4, section 3.3 being the analysis of the individual parameters comparing normal subjects with the groups of suspected glaucoma and confirmed glaucoma.
Author Response
El título solo hace referencia al campo visual y no al OCT que también se incluye en el estudio.
Acordado. Hemos cambiado el título a: "Análisis colorimétrico totalmente automatizado del nervio óptico con la ayuda de Deep Learning y su asociación con la perimetría y la OCT para el estudio del glaucoma".
Línea 40. El examen de la cabeza del nervio óptico también se ha utilizado durante muchos años antes de que se introdujeran otros análisis estructurales.
Cierto. Hemos cambiado la frase "Durante muchos años, la evaluación del campo visual y el examen de la cabeza del nervio óptico fueron los principales procedimientos de diagnóstico".
Línea 62. Angiografía OCT
Lo hemos corregido. Muchas gracias por verlo.
Línea 96. Restante 55%
¡Por supuesto! Gracias por detectar el error.
Line 98. It would be interesting to know more about how the inner edge of Elschnig's scleral ring is delimited.
As this is a methodological issue, the explanation in section A of the appendix "Computing development setup" has been improved as follows: “This was done by manually identifying by an expert. Elschnig's scleral ring is observed as a thin white ring, immediately adjacent to the margin of the optic disc. An example is shown in figure 1 of reference 13.” A copy of this paper is attached.
Line 167. The classification criteria for the groups suspected of glaucoma and confirmed glaucoma are not clear in the text. Please clarify to confirm that there is no selection bias. A table with the different criteria used would be useful for readers.
We believe that the criteria for separating the two groups was clearly defined and in accordance with standard criteria in the literature. Suspected glaucoma was defined from line 173 onwards: "...., such as intraocular pressure greater than 21 mmHg, associated with a family history of glaucoma, an optic disc with a dubious appearance, or borderline visual fields, such as a mean deviation exceeding -2 dB or points outside normal limits on the defect curve. Subjects with ocular pressures greater than 25 mmHg or pressures between 21 and 25 mmHg accompanied by a thin cornea (less than 500 μm) were also included.” Glaucoma was considered to be confirmed when, in addition to the presence of risk factors, the following were observed “the presence of glaucomatous visual field loss in standard automated perimetry (pattern standard deviation or mean deviation of <5%) and signs of glaucomatous neuropathy (Laguna ONhE GDF or OCT-rim area of <5%). Those who did not meet these criteria but met some of the criteria described in the pre-vious paragraph were considered as glaucoma suspects.” (2.3 last paragraph)
We understand that our criterion of not separating confirmed and suspected glaucoma may not be the most common in the literature, but it is the result of a lifetime devoted to this problem. We do so as not to introduce bias by choosing a gold standard “a priori” that may be erroneous. When comparing the diagnostic ability of various procedures in this type of sample, the one that provides the highest sensitivity for high specificity should be the one that performs best. However, as we know this is not the prevailing trend in the current literature, “Two types of analysis were carried out. In the first, no strict boundary was established between confirmed and suspected glaucoma, so as not to introduce an a priori criterion that could alter an objective interpretation of the results… Additionally, two groups were analysed separately” We have added in line 181 “Additionally, two groups were analysed separately (confirmed glaucomas and glaucoma suspects): …”
The key issue is whether we are obliged to reproduce the conventional view, or whether it is permissible for a researcher to have his/her own individual, reasonable criteria, even if they do not agree with the mainstream or the reviewer's views. We believe that there are currently no clear lines that separate these groups, and that the reason for papers such as this one is precisely to help define them. Bias occurs when strict a priori criteria are established to separate suspected from confirmed glaucoma or in the selection of suspected cases.
Recomendamos amablemente al revisor que lea la descripción detallada de nuestro enfoque descrito en la referencia 23. Para ayudar más al revisor, adjuntamos el documento citado. Sería muy complejo describirlo en detalle en el texto escrito de este documento.
Pero insistimos, aunque nuestro criterio es comparar varios procedimientos diagnósticos sin separar la sospecha de la confirmación "a priori", también hemos incluido su análisis por separado, de acuerdo con el enfoque predominante en la literatura, enfoque con el que no estamos del todo de acuerdo. .
Línea 287. Debe ser la sección 3.4, siendo la sección 3.3 el análisis de los parámetros individuales comparando sujetos normales con los grupos de glaucoma sospechado y glaucoma confirmado.
Aunque, como se explicó en la respuesta anterior, no lo consideramos necesario, hemos incluido una nueva sección (3.3) con una nueva tabla y hemos vuelto a numerar la antigua 3.3 a 3.4 en respuesta a la solicitud del revisor.
Reviewer 2 Report
Gonzales-Hernandez et al. reported the diagnostic ability of Laguna ONhE in comparison with other parameters from glaucoma-diagnostics. The authors have investigated extensively to confirm the glaucoma-diagnostic ability of Laguna ONhE related parameters, but the revision in the composition of study population and selection of parameters for comparison should be considered. It is well known that the diagnostic ability is greatly affected by the disease severity. Overall sensitivities seem to be too low at both 95% and 99% specificities and this may be attributable to mixture of “evident glaucoma” and “ambiguous glaucoma suspects”. The definition of “glaucoma suspects” in the present study is too ambiguous and indeed, it raises following questions: (1) are they really patients at high risk of glaucoma (e.g. family history of glaucoma, IOP >21 mmHg, borderline VFs)?; (2) are they suitable to represent glaucoma (or glaucoma-suspect) population that needs high suspicion and early diagnosis? Separate analysis of definite glaucoma and glaucoma suspects may be helpful. Also for the definite glaucoma population, patients with more diverse severity should be included to overcome the limitation of current results mainly affected by advanced glaucoma patients (mean MD -12.16 dB). A large proportion of patients at early stage should be included to support their conclusion of Laguna ONhE providing early sensitivity.
Author Response
There are three aspects of the reviewer's comments that need to be considered:
The first is that "The definition of "glaucoma suspects" in the present study is too ambiguous". We have responded to this in the second last response to reviewer 1, which we will repeat here:
We understand that our criterion of not separating confirmed and suspected glaucoma may not be the most common in the literature, but it is the result of a lifetime devoted to this problem. We do so as not to introduce bias by choosing a gold standard “a priori” that may be erroneous. When comparing the diagnostic ability of various procedures in this type of sample, the one that provides the highest sensitivity for high specificity should be the one that performs best. However, as we know this is not the prevailing trend in the current literature, “Two types of analysis were carried out. In the first, no strict boundary was established between confirmed and suspected glaucoma, so as not to introduce an a priori criterion that could alter an objective interpretation of the results… Additionally, two groups were analysed separately….” We have added “Additionally, two groups were analysed separately (confirmed glaucomas and glaucoma suspects)…”
The key issue is whether we are obliged to reproduce the conventional view, or whether it is permissible for a researcher to have his/her own individual, reasonable criteria, even if they do not agree with the mainstream or the reviewer's views. We believe that there are currently no clear lines that separate these groups, and that the reason for papers such as this one is precisely to help define them. Bias occurs when strict a priori criteria are established to separate suspected from confirmed glaucoma or in the selection of suspected cases.
We kindly encourage the reviewer to read the extensive description of our approach described in reference 23. To further assist the reviewer, we attach the cited paper. It would be very complex to describe in detail in the paper text of this paper.
But we insist, although our criterion is to compare various diagnostic procedures without separating suspicion from confirmation "a priori", we have also included their analysis separately, in accordance with the prevailing approach in the literature, an approach that we do not fully agree with.
The second is that the reviewer states that "Separate analysis of definite glaucoma and glaucoma suspects may be helpful". This criticism is incomprehensible, because this is precisely what has been done in this study. Both populations have been analyzed as a whole, without establishing a specific limit, since such a limit is not precisely known, and they have also been studied separately using a criterion common in the literature (“the presence of glaucomatous visual field loss in standard automated perimetry (pattern standard deviation or mean deviation of <5%) and signs of glaucomatous neuropathy (Laguna ONhE GDF or OCT-rim area of <5%”). Although we do not consider it necessary, we have included a new section (3.3) with a new table and renumbered it from 3.3 to 3.4 in response to the reviewer's 1 request.
And the third one is not easy to understand either because the reviewer says that "Also for the definite glaucoma population, patients with more diverse severity should be included to overcome the limitation of current results mainly affected by advanced glaucoma patients (mean MD -12.16 dB)" But in fact, the reviewer focuses exclusively on the mean but not on the standard deviation (± 7.81 dB). Confirmed glaucomas cover the whole range of the disease, from MD=-0.27 to MD=-31.29. There are not few cases with early defects: 79 cases have MD>-6dB. They can be consulted by the reviewer in the repository we have provided at the end of the paper. https://cloud.insoft.es/s/sRaFgmjQp7S4sgR.
It is obvious that if the sample only contained cases with deep defects, the sensitivity values would be high. The low sensitivity values for the sample as a whole are caused by the fact that we have not been strict in defining suspected glaucoma (Line 179)
Round 2
Reviewer 2 Report
Revised version of the manuscript is accepted.
This manuscript is a resubmission of an earlier submission. The following is a list of the peer review reports and author responses from that submission.
Round 1
Reviewer 1 Report
The authors studied the automatic analysis for Laguna ONhE method in this study. Automation of analysis is essential. I think this paper is methodological, but for some reason, the deep learning part is in the appendix. I cannot discuss the results because I have significant concerns about building models for deep learning.
Major
- If you are using image data, you need to specify the size and resize of the input image.
- Did you use data augmentation? If you are not using it, then you have too little data.
- The author describes the top three layers that use SGD after GAP. Usually, SGD (maybe momentum) uses an optimizer for optimizing the network loss, not the activation function. Why did the authors use SGD for activation function? Moreover, the dropout and/or batch normalization layers between fully connected layers are commonly used to prevent overfitting.
- I strongly recommend that the authors confirm the result of inference using Grad-CAM etc. The accuracy alone does not tell whether the model is doing what it is intended to do.
Author Response
Fully automated colorimetric analysis of the optic nerve aided by Deep Learning and its association with perimetry for the study of glaucoma
Marta Gonzalez-Hernandez, Daniel Gonzalez-Hernandez, Daniel Perez-Barbudo, Paloma Rodriguez-Esteve, Nisamar Betancor-Caro and Manuel Gonzalez de la Rosa
The authors studied the automatic analysis for Laguna ONhE method in this study. Automation of analysis is essential. I think this paper is methodological, but for some reason, the deep learning part is in the appendix. I cannot discuss the results because I have significant concerns about building models for deep learning.
We agree with the reviewer, automation of the analysis is essential, especially for reproducibility purposes. We also agree that the article has an important methodological basis, but its objective is clinical and it is to be published in a clinical medical journal. That is why the deep learning part was added as an appendix, so that the medical reader is not overwhelmed by the description. However, if the editors consider it appropriate, the entire appendix may be relocated as an additional section to Materials and Method.
Major
- If you are using image data, you need to specify the size and resize of the input image.
Since the reviewer does not specify to which of the 5 networks are the comments on, we have assumed that all points refer to Normal vs Glaucoma (E).
The following comments have been added to the text: “Therefore, independently of the original images size, which was 1956x1934, all were trimmed around the ONH segmentation and resized to 224x224 in order to fine-tune the pretrained ResNet50 network.”
- Did you use data augmentation? If you are not using it, then you have too little data.
Data augmentation was used, even though transfer learning allows for using smaller sets of data. In order to further avoid overfitting, random horizontal flip, rotation of up to 25º and brightness darkening up to a 20%, was implemented. No other data augmentation methods were used that could compromise the shape or position of the optic nerve. These clarifications have been added to the text.
- The author describes the top three layers that use SGD after GAP. Usually, SGD (maybe momentum) uses an optimizer for optimizing the network loss, not the activation function. Why did the authors use SGD for activation function? Moreover, the dropout and/or batch normalization layers between fully connected layers are commonly used to prevent overfitting.
The reviewer is right in that a more precise description of the network may be helpful. After each of the first two fully connected layers, a dropout layer with a rate value of 0.5 was included. Also, both use a Hyperbolic Tangent (tanh) activation function. The final one uses a softmax activation function. Categorical cross-entropy was used as loss function and accuracy was used as a metric to judge the performance of the model. Stochastical Gradient Decent (SGD) was used as an optimizer with learning rate 0.0001 and momentum 0.9. These clarifications have been added to the text.
- I strongly recommend that the authors confirm the result of inference using Grad-CAM etc. The accuracy alone does not tell whether the model is doing what it is intended to do.
Indeed, in our usual way of working we continuously confirm the results of the inferences, using the Learning Curve and Confusion Matrix. We find them more useful than Grad-CAM, because we agree with Adebayo's el al opinion that “ visual inspection of explanations alone can favor methods that may provide compelling pictures, but lack sensitivity to the model and the data generating process” (Adebayo, J.; Gilmer, J.; Muelly, M.; Goodfellow, I.; Hardt]y, M.; Kim, B.; Sanity Checks for Saliency Maps. 32nd Conference on Neural Information Processing Systems (NeurIPS 2018), Montréal, Canada).
We expose the results in the new figures 5 a and b, which have been included in the appendix with the following explanation:
In order to assess the model’s efficiency [42], the Learning Curve of the training and validation loss as well as the confusion Matrix of the test dataset are shown in Fig 5a and Fig 5b. These graphical representations are the most commonly accepted methods to show the model’s accuracy as well as how well it generalizes [43, 44].
As can be seen from the learning curve, we have achieved an optimal fit which is the goal of any learning algorithm. As it is desired in any good machine learning algorithm, the generalization gap has been reached which is minimal [45]. Both the training and validation losses decreased to a point of stability. After 1000 Epochs the training was stopped in order to avoid overfitting. The confusion matrix showed that the model had very good accuracy on the test data-set.
Reviewer 2 Report
- lots of jargon makes abstract difficult to read - I suggest a bit of rewording
- using colour of ONH as indicator of perfusion which authors have previously shown to have good sensitivity/specificity
- the team has done a lot of work on Laguna ONhE previously and is now applying AI to allow automation & compare to VFs/OCT
- how does this work in aspects of ONH pallor whether due to other conditions or with cataract surgery where replacing the natural lens can cause the retina to appear paler? Is there a sufficient dataset to train on this?
- is there an indication of how accurate the segmentation of the optic disc and boundaries are? was a comparison made between trained image graders and with OCT scans through the ONH. It appears in Fig 1b that the boundary of the optic disc is misjudged (too small) and the cup is too large, in particular inferiorly
- do the training images cover a sufficiently diverse population, sex, ethnicity, age, refractive error?
- Table 1, some of this is bound to have good AUC based on how a participant was labelled as 'glaucoma' group - if they had a dubious disc appearance, it is likely that their OCT rim area/vertical C/D would be an outlier - meaning you are bound to have good AUC. And by not necessarily having a VF defect, then MD may not score high as you can have a more normal MD and still be classed as ‘glaucoma’
- Fig 4 values are difficult to read, please enlarge
- please don’t switch between r and R2 in text and in Fig 4, it’s confusing to use 2 sets of numbers
- in confirmed glaucoma cases, clinically we would look at the sectoral changes to RNFLT too and not just general RNFLT, how does this affect the AUC?
- given the GDF has such high sensitivity and specificity, are the authors suggesting that it is always a (lack of) perfusion issue in early glaucoma?
- what types of glaucoma are in your cohort?
- clinically, our ability to diagnose obvious glaucoma is already quite good, it is more the suspects that we have difficulty with - do you aim to follow these people to see which of these convert to confirmed? or do you aim to look at risk of progression? this is, I would argue, an important step as this is what is missing in our clinic
Author Response
- Lots of jargon makes abstract difficult to read - I suggest a bit of rewording.
We have tried to make the abstract more understandable for the readers.
- Using colour of ONH as indicator of perfusion which authors have previously shown to have good sensitivity/specificity.
- The team has done a lot of work on Laguna ONhE previously and is now applying AI to allow automation & compare to VFs/OCT.
We appreciate comments 2 and 3. This work verifies the previous ones in a larger sample and tries not only to compare the method with VFs/OCT, but also to evaluate their combination.
- How does this work in aspects of ONH pallor whether due to other conditions or with cataract surgery where replacing the natural lens can cause the retina to appear paler? Is there a sufficient dataset to train on this?
Indeed, the absorption of the lens, the colour response characteristics of the fundus camera and the spectral composition of the flash modify the colour of the image. The solution to this problem is to use an intraocular chromatic pattern, which is the colour of the blood in the veins and arteries of the optic nerve. This way, haemoglobin values relative to this pattern are obtained. This has been explained in detail in our first paper (reference 4) and has therefore not been insisted upon. In addition, we have published that the colour of the blood vessels can be used to evaluate the ageing state of the lens (Gonzalez-de-la-Rosa, M.; Gonzalez-Hernandez, M.; Rodriguez-Esteve, P.; Mendez-Hernandez, C.; Preliminary results of a new method for measuring the spectral absorption of the crystalline lens in vivo. J Cataract Refract Surg 2018, 44, 512-513).
- Is there an indication of how accurate the segmentation of the optic disc and boundaries are? was a comparison made between trained image graders and with OCT scans through the ONH. It appears in Fig 1b that the boundary of the optic disc is misjudged (too small) and the cup is too large, in particular inferiorly.
In a previous publication (reference 10) we used the Laguna ONhE method by segmenting the photographic images with an overlapped OCT image. But to automate the method it had to be done independently of external instruments. Therefore, the training for the segmentation was based on the subjective segmentation by a single expert of 40,000 optical discs. This is referred to in section A of the appendix to this paper, citing the original publication (reference 13) which can be obtained from the following address:
https://www.semanticscholar.org/paper/Segmentation-of-the-Optic-Nerve-Head-Based-on-Deep-Gonzalez-Hernandez-Diaz-Aleman/6e19ccb60e676e86fdb0f3d7a5f5254b0c2f5375
The reviewer's impression that the segmentation of the nerve boundaries presented in Figure 1 is too internal is due to the criterion used. The apparent (clear) edge is not delineated, but rather the inner edge of Elschnig's scleral ring. The following sentence was added to the manuscript: ...”the inner edge of Elschnig's scleral ring is defined, which is more internal than the apparent clear edge, and the size and shape of the cup is estimated."
The size and shape of the cup may also differ from that estimated by stereoscopic photography or various OCTs, as they use different criteria. In our case they are calculated by multiple correlation using the amounts of relative haemoglobin in multiple sectors of the disc, as explained in another paper (reference 5) and may differ somewhat from the interpretation of each observer, especially when looking at a two-dimensional image.
- O the training images cover a sufficiently diverse population, sex, ethnicity, age, refractive error?
The set of training images is big enough, so that no relevant influences of age or gender are to be expected. Refractive defects greater than 5 dioptres of spherical equivalent have not been excluded from training, but are excluded from the evaluation of the results. The influence of high refractions will have to be assessed in specific studies. The influence of disc size on the results is also currently under study.
Most of our samples correspond to European ethnic groups and to a lesser extent to African and American ethnic groups. We are currently improving our training, with 400,000 optic nerves available, but mostly subjects originating from these regions. We have little experience with Asian ethnic groups.
- Table 1, some of this is bound to have good AUC based on how a participant was labelled as 'glaucoma' group - if they had a dubious disc appearance, it is likely that their OCT rim area/vertical C/D would be an outlier - meaning you are bound to have good AUC. And by not necessarily having a VF defect, then MD may not score high as you can have a more normal MD and still be classed as ‘glaucoma’
Of course, the suspicion of glaucoma is not based on a single sign, but on the coincidence of several signs. However, the borderline between normality and pathology is not a perfectly defined line and this can always lead to incorrect classifications. For this reason, we have always preferred to study the population of confirmed and suspected glaucomas as a single group. We have defended such criterion with logical arguments in the article cited in reference 23 (see also reference 22). However, we know very well from experience that the scientific habit or convention of making this separation into groups cannot be avoided, as it is expected by readers, reviewers and editors alike.
- Fig 4 values are difficult to read, please enlarge.
The size of the characters in figure 4 has been increased.
- Lease don’t switch between r and R2 in text and in Fig 4, it’s confusing to use 2 sets of numbers.
The value of R2 has been removed from the graph and retained r in the text.
- In confirmed glaucoma cases, clinically we would look at the sectoral changes to RNFLT too and not just general RNFLT, how does this affect the AUC?
In all honesty, it is impossible to analyse all the variables provided by the OCTs or Laguna ONhE in a paper like this. Particularly the RNFLT sectors, some may be normal and some not, and it is difficult to decide which to present (all of them? the minimum number?). We already made a sectorial division of the OCT- RNFLT in our first Laguna ONhE paper (reference 4). Laguna OHnE also calculates the areas and amounts of relative haemoglobin at the cup and at six sectors of the rim, but we have not included it in this work either. We feel than doing so would make the paper even more complex than it already is.
- Given the GDF has such high sensitivity and specificity, are the authors suggesting that it is always a (lack of) perfusion issue in early glaucoma?
This is not just our suggestion. It is suggested by many authors. This is what we address in the last sentence of the introduction. But the word "always" would imply a certainty that we are not currently in a position to sustain.
- What types of glaucoma are in your cohort?
The first paragraph of section 2.3 states that that the cohort consisted of "chronic open-angle glaucoma". The second paragraph of the first version states that "open anterior chamber angle" was checked.
- Clinically, our ability to diagnose obvious glaucoma is already quite good, it is more the suspects that we have difficulty with - do you aim to follow these people to see which of these convert to confirmed? or do you aim to look at risk of progression? this is, I would argue, an important step as this is what is missing in our clinic.
You are absolutely right. Often 'reliable early diagnosis' lies in the observation of progression in data that is close to normal. Therefore, isolated values are not always reliable.
In fact, what we are currently looking for is a better method for the analysis of progression, both in early, moderate and advanced glaucoma. Therefore, in our opinion, the most relevant aspect of this work is the linearity observed in the association between perimetric MD and the GDF-TCV (Figure 3a). It is widely accepted that morphological defects may precede functional defects, but also that in deep defects perimetry has a wider rank than morphology. Laguna ONhE associated with visual field irregularity is likely to allow estimating the depth of defects in advanced stages in a linear fashion relative to MD. The latest version of the Laguna ONhE method currently allows progression studies to be carried out and we are planning two such trials.

Reviewer 3 Report
This study is about using previously developed Laguna-ONhE to detect optic nerve head perfusion and comparing the results by analyzing association with data obtained from perimetry using deep learning methods. Methodology, using deep learning, to analyze data is quite intereting, but I do not see the merit of using data from Lagnua-ONhE instead of using data obtained from OCTA, which is currently most widely used method for detecting optic nerve head perfusion and peripapillary retina. Followings must be improved in this manuscript:
- Abstract: it is not clear what GDF means when just looking at abstract. Please describe. Also further description is needed for which indices were used for OCT.
- Overall English grammar check is needed throughout the paper.
- Usually we use term "OCT angiography" or "OCTA", not Angio-OCT.
- What does "nerve" mean? Does author mean by "optic nerve"? Please revise.
- Figure 1. I do not understand what is displayed in figure 1. Authors should more descriptive about the figure in the text as well as in figure caption, especially what is shown in figure 1c and d.
- Table 1-3. Description for Table 1 to 3 is too briefly written in the text. Authors should be more descriptive about what is shown in Table 1-3 for readers' better understanding.
- Discussion. paragraph 1 and 2 should be combined.
- Authors should explain about benefits of using Laguna-ONhE results compared to using data from OCT angiography, which is more recently introduced and more widely used worldwide.
Author Response
This study is about using previously developed Laguna-ONhE to detect optic nerve head perfusion and comparing the results by analyzing association with data obtained from perimetry using deep learning methods. Methodology, using deep learning, to analyze data is quite intereting, but I do not see the merit of using data from Lagnua-ONhE instead of using data obtained from OCTA, which is currently most widely used method for detecting optic nerve head perfusion and peripapillary retina.
The “old” (retinographies) is not always worse than the “new” (OCTA), when the “old” is approached in a completely new way. If it proves no worse or even better than the “new”, the result will not only be relevant, but will naturally surprise users. On the other hand, optic nerve colour retinography can nowadays be performed even with a mobile phone. The implication of telemedicine analysis of these images compared to OCTA images, that can only be taken in specialized care centres, can obviously be very relevant and of great cost-benefit health importance for public health services. A paragraph has been added to the end of the discussion, also addressing point 8.
Followings must be improved in this manuscript:
- Abstract: it is not clear what GDF means when just looking at abstract. Please describe. Also further description is needed for which indices were used for OCT.
We have completely restructured the Abstract and we have added the following sentence: “The best combination of OCT and perimetry was obtained with the vertical cup/disc ratio …”
- Overall English grammar check is needed throughout the paper.
The new version has been reviewed by one of the companies recommended by JCM for English language editing. A certificate is attached.
- Usually we use term "OCT angiography" or "OCTA", not Angio-OCT.
The text has been corrected accordingly.
- What does "nerve" mean? Does author mean by "optic nerve"? Please revise.
The text has been corrected accordingly.
- Figure 1. I do not understand what is displayed in figure 1. Authors should more descriptive about the figure in the text as well as in figure caption, especially what is shown in figure 1c and d.
The following explanation has been added before displaying the figure: “It shows how the method automatically performs the analysis of a retinography (a). Once its quality has been checked, the position of the optic nerve is identified, the inner edge of Elschnig's scleral ring is defined, which is more internal than the apparent clear edge, and the size and shape of the cup is estimated (b). Its veins and arteries are then segmented, the colour of which is used to estimate the relative haemoglobin of the rest of the nerve tissue, shown in a colour code (c). In each sector of the optic disc, cup or rim, their areas are estimated as a percentage of the total disk area, and expressed in colour if it corresponds to what is expected in a normal optic nerve (d)”.
Figure 1 has been corrected because the letter "d" had been displaced in the previous version.
- Table 1-3. Description for Table 1 to 3 is too briefly written in the text. Authors should be more descriptive about what is shown in Table 1-3 for readers' better understanding.
The following comments have been added:
To table 1: " The best OCT index was rim area and the best perimetric index TCV, with no significant differences between them.”
To table 2: "The RNFLT and PSDr combination performed better than the other OCT indexes combi-nations with the perimetric PSDr. The combinations with the perimetric MD performed worse and thus were ignored."
To table 3. “In suspected glaucomas, the combinations of the three OCT indices and the perimetric PSDr did not show significant differences. The combinations with the perimetric MD were worse and thus ignored". In this table, a defect in a heading has also been corrected.”
- paragraph 1 and 2 should be combined.
Paragraphs 1 and 2 of the Discussion have been combined.
- Authors should explain about benefits of using Laguna-ONhE results compared to using data from OCT angiography, which is more recently introduced and more widely used worldwide.
This paragraph has been added to the end of the discussion:
“The combination of the indices provided by OCT angiography with perimetric indices should be evaluated in the future, in order to compare their results with those presented in this paper. However, an important factor to take into account in this type of comparison is the cost–benefit ratio for healthcare systems, given the low cost of some current fundus cameras, and the possibility of evaluating their images via telemedicine, without additional equipment. Such an evaluation should also be carried out in the near future.”
Round 2
Reviewer 1 Report
The authors have made efforts to improve the paper. However, the concerns about deep learning could not be eliminated.
Did the authors confirm the latest paper by Adebayo J? NeurIPS is one of the top conferences in computer vision. However, the period of turnover in computer vision is fast in comparison to other categories. In 2019, Adebayo J published on arXiv for the importance of explanation in artificial intelligence (AI). There is now a need for explainable AI to eliminate the black box of the model. In scientific papers, the performance of a model that analyses the image data requires accuracy and loss and requires the result of visualization.
The authors added the information for training and validation of each epoch in Figure 5. The authors responded that they conducted a data augmentation to prevent overfitting. However, the validation loss has stopped from approximately 100 epochs. Therefore, this model has not been updated in nearly 900 epochs.